# Transcriptomic Analysis of Mating Responses in *Bemisia tabaci* MED Females

**DOI:** 10.3390/insects11050308

**Published:** 2020-05-14

**Authors:** Zhijia Huo, Yating Liu, Jinjian Yang, Wen Xie, Shaoli Wang, Qingjun Wu, Xuguo Zhou, Baoping Pang, Youjun Zhang

**Affiliations:** 1Research Center for Grassland Entomology, Inner Mongolia Agricultural University, Hohhot 010020, China; huozhijianmg@126.com; 2Department of Plant Protection, Institute of Vegetables and Flowers, Chinese Academy of Agricultural Sciences, Beijing 100081, China; liuyating19900209@163.com (Y.L.); yangjj0707@foxmail.com (J.Y.); xiewen@caas.cn (W.X.); wangshaoli@caas.cn (S.W.); wuqingjun@caas.cn (Q.W.); 3Department of Entomology, University of Kentucky, Lexington, KY 40546-0091, USA; xuguozhou@uky.edu

**Keywords:** *Bemisia tabaci* MED, transcriptome, mating responses, DEG, female

## Abstract

Mating triggers substantial changes in gene expression and leads to subsequent physiological and behavioral modifications. However, postmating transcriptomic changes responding to mating have not yet been fully understood. Here, we carried out RNA sequencing (RNAseq) analysis in the sweet potato whitefly, *Bemisia tabaci* MED, to identify genes in females in response to mating. We compared mRNA expression in virgin and mated females at 24 h. As a result, 434 differentially expressed gene transcripts (DEGs) were identified between the mated and unmated groups, including 331 up- and 103 down-regulated. Gene Ontology (GO) and Kyoto Encyclopedia of Genes and Genomes (KEGG) enrichment analyses revealed that many of these DEGs encode binding-related proteins and genes associated with longevity. An RT-qPCR validation study was consistent with our transcriptomic analysis (14/15). Specifically, expression of P450s (*Cyp18a1* and *Cyp4g68*), ubiquitin-protein ligases (*UBR5* and *RNF123*), *Hsps* (*Hsp68* and *Hsf*), carboxylase (*ACC-2*), facilitated trehalose transporters (*Tret1-2*), transcription factor (*phtf*), and serine-protein kinase (*TLK2*) were significantly elevated in mated females throughout seven assay days. These combined results offer a glimpe of postmating molecular modifications to facilitate reproduction in *B. tabaci* females.

## 1. Introduction

Mating plays a pivotal role in the evolution, development and sex-ratio of species, as it enhances biodiversity and maintains reproduction rates. Mating makes females get sperm. Along with sperm, male accessory gland proteins and microorganists are delivered to females [1,2,3,4,5,6,7,8,9], which causes dramatic changes in physiology and behavior to females. For example, mating induces the spermatheca to produce phospholipids, carbohydrates, and proteins that may help maintain sperm viability and ensure the success rate of fertilization in *Drosophila melanogaster* [8]. In addition, mating can increase egg development, oviposition rates and mating refractoriness in *Aedes aegypti* [10,11,12,13]. Mating also triggers substantial changes in gene expression, and such changes have been studied in *D. melanogaster*, *Apis mellifera*, *Anopheles gambiae*, *A. aegypti* and *Ceratitis capitata* [2,3,4,5,6,7]. In *D. melanogaster*, 432 transcripts were differentially expressed in mated females relative to virgin females. Many of these genes encode proteins with predicted functions in catalytic activity and nucleic acid binding. Immune response genes (*cecA1* and *att A*) also showed a substantial increase in expression in response to mating [8]. In *A. gambiae* and *A. aegypti*, differential expression was concentrated in genes that encode various proteases like matrix metalloproteinase [7]. In addition, genes involved in metabolic processes were significantly upregulated in *A. gambiae*, and genes associated with the immune system and antimicrobial function were upregulated in *A. aegypti* [7,13,14]. Most of these changes are conducive to mating success and the continuation of populations [1,2,3,4,5,6,7,8,13,14].

The sweetpotato whitefly, *Bemisia tabaci* MED (Gennadius) (Hemiptera: Aleyrodidae), a global invasive insect pest, not only damages crops and horticultural plants by feeding and secreting honeydew to decrease photosynthesis, but also transmits more than 300 plant viruses [15,16]. Females play a key role in *B. tabaci* outbreaks, because only mated females can produce female offspring. In *B. tabaci*, mating behavior has been studied in the B-type and Asia II groups [17]. Additionally, studies have shown that the global invasion and displacement of *B. tabaci* are associated with mating [18]. By analyzing the postmating transcriptome changes, we intended to identify specific genes and pathways associated with *B. tabaci* reproductive biology. It is our hope that by interfering with the expression of these key genes/pathways, we can control this invasive pest through reduced female offspring. In this study, based on the comparative analyses of six *B. tabaci* transcriptomes between virgin and mated females, we identified the mating-induced changes in *B. tabaci* females. Furthermore, genes putatively involved in mating at different time points were investigated by quantitative reverse-transcriptase PCR (RT-qPCR) analysis.

## 2. Materials and Methods

### 2.1. Insect Rearing and Sample Preparation

The *B. tabaci* population was originally obtained from poinsettia plants (*Euphorbia pulcherrima* Willd. ex Klotz.) in Beijing, China, in 2009. This population was then maintained on cotton in a glasshouse under natural light. Before sample collection, the purity of the strain was confirmed via an mtDNA COI marker [19].

For sample collection, newly emerged females and males (within 1 h of emergence to ensure that they did not mate) were separately collected in glass tubes (5.0 × 0.5 cm), and the sex of each individual was determined with a stereomicroscope. Every five virgin females were released into a plastic bottle (5.5 cm diameter, 15 cm height) with a cotton seedling, and five males were added to each plastic bottle for mating experiments. A second set of plastic bottles did not receive males (25 replicates per treatment). After a 24 h mating [17], the mated and virgin groups were collected. Each treatment contained 3 biological replicates (virgin: CKD1-1, CKD1-2, CKD1-3; mated: D1-1, D1-2, D1-3), and each replicate contained 40 individuals. Then they were frozen in liquid nitrogen and stored at −80 °C until RNA extraction.

### 2.2. RNA Extraction, cDNA Library Construction, and Illumina Sequencing

RNA from each sample was extracted with TRIzol reagent according to the manufacturer’s instructions (Invitrogen, Carlsbad, CA, USA), and RNA concentration was assessed using a NanoDrop 2000 (Thermo Scientific, Wilmington, DE, USA). Purity was checked by 1% *w*/*v* agarose gel electrophoresis. Six samples, containing 6 mg total RNA each, were sent to Biomarker (Biomarker Technologies Corporation, Beijing, China) for cDNA library construction and Illumina sequencing (Appendix A).

Sequencing libraries were generated by the NEBNext Ultra™ RNA Library Prep Kit for Illumina (NEB, Ipswich, MA, USA) following the manufacturer’s recommendations, and index codes were added to attribute sequences to each sample. Briefly, mRNA was purified from total RNA using poly-T oligo-attached magnetic beads. Fragmentation was carried out using divalent cations under high temperature in NEBNext First-Strand Synthesis Reaction Buffer (5×). Then, the first- and second-strand cDNA was successfully synthesized. The remaining overhangs were converted into blunt ends via exonuclease/polymerase activities. The short fragments and adapters were linked together, and then suitable fragments were chosen for subsequent PCR amplification. PCR was performed with Phusion High-Fidelity DNA polymerase, universal PCR primers and Index (X) Primer to obtain Index-coded samples. Finally, PCR products were purified (AMPure XP system), and the library quality was assessed on an Agilent Bioanalyzer 2100 system. Index-coded samples were prepared on a cBot Cluster Generation System using the TruSeq PE Cluster Kit v4-cBot-HS (Illumina, San Diego, CA, USA). The library preparations were sequenced on an Illumina platform, and raw data were obtained. The sequencing data generated in this study have been deposited in the Sequence Read Archive (SRA, Birmingham, UK) database under the Bioproject accession number PRJNA559034.

### 2.3. Comparative Analysis and Functional Annotation

The adaptor sequences and low-quality sequences were removed from the raw data to obtain clean reads for further analysis. These clean reads were then mapped to *B. tabaci* MED reference genome sequence [20] through Hisat2 software to obtain the read alignments. The alignments were then assembled with StringTie, which assembles and quantifies the transcripts in each sample. After the initial assembly, the assembled transcripts were merged together by a special StringTie module, which creates a uniform set of transcripts for all samples. The gffcompare program was then used to compare the genes and merge the transcripts with the annotation and report statistics to obtain transcript statistics.

To annotate the pooled assembly transcriptome, we performed a BLAST search against the nonredundant (NR) database in NCBI, Swiss-Prot (http://www.geneontology.org/), Kyoto Encyclopedia of Genes and Genomes (KEGG; http://www.genome.jp/kegg/) and Clusters of Orthologous Groups of Proteins (COG; http://www.ncbi.nlm.nih.gov/COG/) with an E-value ≤ 1 × 10^−5^. Gene Ontology (GO; http://www.geneontology.org/) terms were assigned by Blast2GO through a search of the NR database.

### 2.4. Identification of Mating-Related DEGs and Enrichment Analysis

The quantification of gene expression levels was estimated by fragments per kb of transcript per million fragments mapped (FPKM). FPKM values were used directly to compare gene expression differences between various samples. The “base means” value for identifying DEGs was obtained using the DESeq package. The transcripts with a FDR ≤ 0.05 and the absolute value of the log_2_ fold change ≥ 1 were considered a DEG in this study. Default parameter settings (*p*-value cut-off for false discovery rate 0.001) in the DESeq package were then used for final DEG analysis to generate outputs in the form of a heatmap [21]. In addition, GO enrichment analysis of DEGs was implemented by the GOseq R packages-based Wallenius noncentral hypergeometric distribution [22], and KOBAS software was used to test the statistical enrichment of DEGs in KEGG pathways [23].

### 2.5. RT-qPCR Analysis

Triplicate samples of both mated and virgin females were collected again, snap frozen in liquid nitrogen, and stored at −80 °C. Each treatment was divided into 3 replicate RNA preparations of 40 whiteflies each for the subsequent RT-qPCR analysis. Total RNA was extracted as described above, and first-strand cDNA was prepared using 1 μg of total RNA with the PrimeScript RT Reagent Kit (TaKaRa Biotech, Mountain View, CA, USA). The resulting cDNA was diluted to 0.1 mg/mL for further analysis by RT-qPCR (ABI-Q3) using SuperReal PreMix Plus (SYBR Green) (Tiangen, Beijing, China) according to the manufacturer’s instructions. Each reaction system contained 1 μL of cDNA template, 10 μL of SuperReal PreMix Plus, 0.4 μL of ROX reference dye, 0.6 μL of specific primers and 7.4 μL of ddH_2_O. PCR was performed under the following conditions—denaturation at 95 °C for 10 min, followed by 40 cycles of 95 °C for 15 s, 60 °C for 30 s and 72 °C for 30 s. We selected 15 DEGs for RT-qPCR validation, and some homologies of those genes had been reported in other species of previous mating related transcriptomes documents [1,2,3,4,5,6,7,8]. Carboxylases and hydrolases were unique to the postmating *B. tabaci* females. Specific primers were designed using Primer Premier 5.0 software (Appendix A). Three independent biological replicates were executed for each sample. Data were normalized to the RPL29 gene [24], and relative gene expression was calculated using the 2^−ΔΔCt^ method [25]. SPSS 19.0 was used to analyze correlations between RT-qPCR data and RNA-seq data.

### 2.6. Mating-Related Genes Expression Profiles Analysis at Different Time Points

We collected 1-, 3-, 5- and 7-day mated and virgin females. Each sample was collected in triplicate. RNA was extracted and stored at −80 °C.

We selected all significantly upregulated DEGs which were verified in “materials and methods 2.5”, and analyzed their expression profiles in mated and virgin females at different time points by RT-qPCR.

### 2.7. Statistical Analysis

One-way ANOVA with Tukey’s test (*p* < 0.05) was used to evaluate differences among treatments. Values presented in figures represent the means calculated from biological replicates and their corresponding standard errors.

## 3. Results

### 3.1. Illumina Sequencing and Clean-Read Map

Through transcriptome analysis of the six samples, a total of 41.10 Gb clean data was obtained. The clean data of each sample reached 6.18 Gb and the Q30 base percentage was 92.27% or higher. GC content ranged from 39.16% to 43.90%. The clean reads of each sample were aligned against the *B. tabaci* MED reference genome [20], with an efficiency ranging from 75.51% to 83.52% (Appendix A). A total of 25,594 transcripts were obtained, including 4846 novel genes that were named Bemisia_tabaici_newGene, of which 2510 were functionally annotated (Appendix A).

### 3.2. Functional Annotation and Classification

A total of 25,594 transcripts were searched against eight databases (NR, Swiss-Prot, Pfam, KEGG, COG, GO, euKaryotic Orthologous Groups (KOG) and evolutionary genealogy of genes: Nonsupervised Orthologous Groups (eggNOG)) and annotated in at least one database (Appendix A). Of the 25,594 transcripts, approximately 80% (20,474) could be annotated in NR. Of these 20,474 transcripts, 60.1% were longer than 1000 bp, 36.8% were 300–1000 bp, only 3.1% transcripts were less than 300 bp (Appendix A). Additionally, 4320 transcripts could be annotated into 51 GO terms. Of these 4320 transcripts, genes related to catalytic activity were the most abundant (2199/50.90%), followed by genes related to binding (1953/45.21%) and metabolic processes (1849/42.80%) (Appendix A). Although 8008 transcripts were annotated to the KEGG database, only 3718 transcripts were annotated to 223 KEGG pathways. The analysis revealed that lysosome-related genes were the most abundant for 244 transcripts (3%), then 191 transcripts (2%) were associated with RNA transport (Appendix A).

### 3.3. Statistics Analysis of Mating-Related DEGs

The PCA analysis of samples (Appendix A) showed a satisfactory biological replicate in mated and virgin groups. Based on those, we identified 434 DEGs by comparing mated and virgin females, of which 331 were upregulated and 103 were downregulated (Table 1 and Figure 1). Four novel genes (Bemisia_tabaci_newGene_33448, Bemisia_tabaci_newGene_4341, Bemisia_tabaci_newGene_20113, Bemisia_tabaci_newGene_35238) were unique to the postmating *B. tabaci* females (Appendix A).

### 3.4. GO Annotation and KEGG Pathways Analysis of DEGs

A total of 77 DEGs were annotated in GO terms, of which 66 were upregulated and 11 were downregulated (Table 1). The analysis showed that upregulated DEGs related to binding (32) and catalytic activity (28) were the most abundant in molecular function, followed by cellular processes (28), metabolic processes (27) and single-organism processes (23) in biological process (Figure 2 and Appendix A). In turn, the 11 downregulated DEGs were associated with binding (7), cellular processes (6), and catalytic activity (5) (Figure 2 and Appendix A). Among these GO terms, the cellular component contained the nucleus and intracellular part, and the molecular function contained transcription factor activity, sequence-specific DNA binding, and oxidoreductase activity, acting on the CH-OH group of donors terms were significantly enriched (Appendix A).

The DEGs were also mapped into canonical KEGG pathways to identify possible active biological pathways. In those pathways, most genes were upregulated. Specifically, protein processing in the endoplasmic reticulum, endocytosis, longevity regulating pathway-multiple species, foxo signaling pathway, phosphatidylinositol signaling, pyruvate metabolism and spliceosome contained more than five upregulated DEGs (Appendix A). Regarding downregulated DEGs in pathways, the phagosome pathway contained three DEGs and the spliceosome contained two DEGs. In 11 pathways, each pathway only contained one DEG, with the rest of the pathway without any DEGs (Appendix A). Among these pathways, longevity regulating pathway-multiple species was significantly enriched (Table 2). NR annotation revealed that some of the genes previously reported to be associated with mating have also been found in DEGs, including cytochrome P450, serine-protein kinases (*SPK*), ubiquitin-protein ligase, etc. A detailed description of those DEGs is shown in Appendix A.

Comparison of GO terms with KEGG pathways indicated that 33 DEGs were in common which have played different roles in key pathways that are possibly responsive to mating in *B. tabaci*. Especially, ubiquitin-dependent protein catabolic process, trehalose biosynthetic process, response to stress and lipid metabolic process of biological process in GO terms were related to ubiquitin mediated proteolysis, starch and sucrose metabolism, longevity regulating pathway-multiple species and glycerolipid metabolism pathways, respectively (Appendix A). Regarding molecular function, glucose-6-phosphate dehydrogenase activity and phosphoenolpyruvate carboxykinase activity were associated with pentose phosphate, glycolysis/gluconeogenesis, citrate cycle and pyruvate metabolism pathways (Appendix A).

### 3.5. Real-Time PCR Validation of Mating-Related Genes

In this study, we selected 15 DEGs to confirm the validity of mating-related genes in RNA-seq results. Three ubiquitin-protein ligase genes (*UBR5*, *UHRF1* and *RNF123*), two cytochrome P450 genes (*Cyp4g68* and *Cyp18a1*), two heat-shock protein genes (*Hsp68* and *Hsf*), two facilitated trehalose transporter genes (*Tret1* and *Tret1-2*), two serine-protein kinase genes (*TLK2* and *SBK1*), two carboxylase genes (*ACC-1* and *ACC-2*), transcription factor (*phtf*) and hydrolase (*DDAH-1*). Cytochrome P450 families and *Hsp* families have been reported before [1,2,3,4,5,6,7,8]. However, *Cyp4g68*, *Cyp18a1*, *Hsp68,* and the rest genes were unique to mated *B. tabaci* females. These were chosen for RT-qPCR validation (Appendix A). As observed by RNA-seq and RT-qPCR (Figure 3 and Figure 4), 93.3% of these selected genes had consistent expression, except for ubiquitin-protein ligases *UHRF1*. The correlation analysis showed that RT-qPCR and RNA-seq data were significantly correlative except for *UHRF1* (Appendix A).

### 3.6. Expression Profiles of Mating Related Genes at Different Time Points

A total of 13 mating-related DEGs (verified in result 3.4) were used to verify the expression profiles at different time points within 7 days. *Cyp18a1* was gradually increased for mated females within 7 days, while *UBR5* and *RNF123* showed a downward trend after the first rise (Figure 5). In contrast, the expression profiles of *Hsf* and *TLK2* did not change at different time points (Figure 5). Although the expression level of *Tret1-2* and *phtf* tended to fluctuate, it was still always higher than that of virgin females at different time points (Figure 5).

The expression levels of the remaining genes gradually decreased. However, within 7 days, *ACC-2*, *Hsp68* and *Cyp4g68* were always upregulated in mated females in comparison with virgin females (Figure 5). Interestingly, the expression levels of *SBK1*, *Tret1* and *ACC-1* in mating females decreased sharply after mating for 1 day, and reached levels similar to those of virgin females, especially at 5 and 7 days (Figure 5). For virgin females, most mating-related genes maintained low expression levels at the different time points (Figure 5).

## 4. Discussion

Mating is central to reproductive success in vertebrates and invertebrates [5,7,26,27], and previous studies have shown that mating has profound effects on female biology and behavior [5,6,7]. The whitefly, *B. tabaci* (Hemiptera: Aleyrodidae), is a complex species with a haplodiploid reproductive system [28]. Studies showed that asymmetric mating interactions lead to widespread invasion and displacement of whiteflies [18]. Because only mated females of *B. tabaci* can produce female offspring, we compared transcript abundance levels in virgin females with those of mated females. Six samples were used for library construction and comparative analysis. Though DESeq to comparative transcriptome analysis resulted in 434 candidate mating-related genes, which was a cost-effective strategy [29]. Comparison of these DEGs with other mating related transcriptome results revealed that lipid transport proteins, cytochrome 450 families, immune response genes, transcription factors, heat shock proteins, response to stimulate genes and serine/threonine-protein kinases had similar expression profiles [3,7,8,30]. However, glucose dehydrogenases were upregulated in *B. tabaci* females, and were downregulated in *A. aegypti* [7]. Histones were upregulated in *Drosophila* females, but downregulated in postmating *B. tabaci* (Appendix A) [30]. The number and expression profile of DEGs varies within time points, with some genes maintaining a steady expression trend and others reversing it [7,30]. With the passing of the time, multiple mates induce complex changes, especially in metabolism. Some genes were expressed immediately after mating and maintained their expression profiles subsequently, and some genes had fluctuating expression levels. These may be related to gene function at different time points [3,7,26].

The GO function categories of these DEGs were associated with molecular transducer activity, binding, response to stimulus, and metabolic processes (Appendix A), which is consistent with previous reports in *D. melanogaster*, *A. gasmbiae* and *A. aegypti* [6,7,14]. Other genes, particularly those related to cell, signal transducer activity, and biological adhesion were unique to the postmating *B. tabaci* females, which may cause physiological changes related to fertilization in mated *B. tabaci* females. Enrichment of differentially expressed genes of KEGG pathways showed that longevity regulating pathway-multiple species was the significant enrichment, which was related to longevity. Mating decreased the longevity of mated females by transmitting virus and mating behavior caused damage to females [9,12], so we speculate that mating would decrease the longevity in mated *B. tabaci* females. The accuracy of the corresponding genes that were differentially expressed in the transcriptome was confirmed by RT-qPCR analysis. Among 15 DEGs selected from the transcriptome for validation, one DEG showed inconsistencies between qPCR and RNE-Seq data. This situation also happens in other species [31], and it is likely caused by false-positivity [9].

Mating triggers large changes in gene expression. The response of genes to mating is complex, especially in P450 genes. Insect cytochrome P450 families comprise a diverse class of enzymes involved in detoxification and the biosynthesis of ecdysteroids and juvenile hormones [32]. In *Drosophila*, six cytochrome P450 genes (*Cyp9f3*, *Cyp307a1*, *Cyp315a1*, *Cyp4p3*, *Cyp313a4* and *Cyp6a21*) were upregulated in mated females, while 22 were downregulated [8,30]. In this paper, all cytochrome P450s were upregulated in mated females compared to virgin females at different time points (Figure 5). During the mating process, males introduce slightly toxic seminal fluid and pathogens into females, which changes the microenvironment of spermatheca, and the upregulated P450 genes may take part in detoxification to ensure successful fertilization [33,34,35]. In addition, the role of P450s in the biosynthesis of juvenile hormones and ecdysteroids could help regulate hormone levels after mating [30]. Specifically, *Cyp18a1* was significantly upregulated and its expression levels increased steadily in mated females (Figure 5). *Cyp18a1* shows a positive response to mating because it was annotated in insect hormone biosynthesis (Appendix A) and played a key role in hormone metabolism due to the increase of hormone synthesis after mating [36].

Among insects, heat-shock proteins are synthesized and induced by environmental and genetic stressors. They act as molecular chaperones to help organisms cope with different kinds of stresses in biological process to improve species survival and population development [37,38]. In this study, mating led to the upregulation of *Hsp68* and *Hsp70* genes at different time points in mated females. Other members of *Hsp* families were also found in mated *A. mellifera* and *Drosophila* females [2,8,26]. With the increase of mating times, the upregulation of *Hsp* genes likely contributes to resistance to changes in the female microenvironment and to maintaining sperm activity to successful mating and fertilization [38,39].

Successful mating and fertilization require energy [40,41]. The energy metabolism pathway was activated after mating in this study. Trehalose is the major blood sugar in insects and an instant source of energy [42]. Trehalose transporters can facilitate trehalose transfer and maintain haemolymph trehalose levels in insects [43,44]. *Tret1-2* may be more important than *Tret1* because the expression of *Tret1-2* was always higher in mated females within 7 days (Figure 5). In addition, two maltase genes (BTA000413.1.gene and BTA022129.2.gene) were also upregulated in mated females (Appendix A), which also help digestion in order to get energy [45]. In mated females, ubiquitin ligases encoded by *UBR5* and *RNF123* were similar to *Tret1-2* in terms of expression level. Ubiquitin ligases have been reported to be essential for substrate recognition and ubiquitination and contribute to supporting the next identification, maintenance, and modification of gametes [46,47], which are beneficial to successful fertilization. Pheromones play an important role in mating by promoting mate attraction and selection, alertness, defences and aggregation [48,49,50]. Carboxylases are involved in the production and biosynthesis of pheromones [51,52], therefore, the upregulation of carboxylases (*ACC-2*) in mated females within 7 days may attract males to increase the success rate of mating.

Mating usually induces upregulation of immune response-related genes [8,30]. We demonstrated two upregulated heparan sulfate genes (BTA014754.1.gene and BTA017216.1.gene) that were ubiquitous glycosaminoglycan with multiple roles in immunity [53]. Similarly, four serine-protein kinases (BTA024078.1.gene, BTA007976.1.gene, *TLK2* and *SBK1*), two retrovirus-related Pol polyproteins (BTA016721.2gene and BTA004002.2gene), and two *Hsp70* genes (BTA003886.2.gene and Bemisia_tabaci_newGene_18814) were obtained from DEGs, and these types genes have been reported to be related to immunity response in pearl oyster [54,55,56,57]. They may contribute to the defense against invading microbes transmitted by mating to ensure successful mating. All of these key genes discussed above might play different roles in postmating females to ensure successful fertilization. Next, their functions in *B. tabaci* need further investigation.

## 5. Conclusions

Undoubtedly, successful mating is essential to the survival of all species. This study provides a detailed understanding of the mating-related genes in *B. tabaci*. The mating process causes some genes to be overexpressed. The overexpressed genes described here play different roles in the sophisticated mating process by redistributing resources from somatic maintenance to mating processes and microenvironmental protection. Finally, 10 genes were upregulated in mated females at different time points, which may play key roles in mating.

## Figures and Tables

**Figure 1 insects-11-00308-f001:**
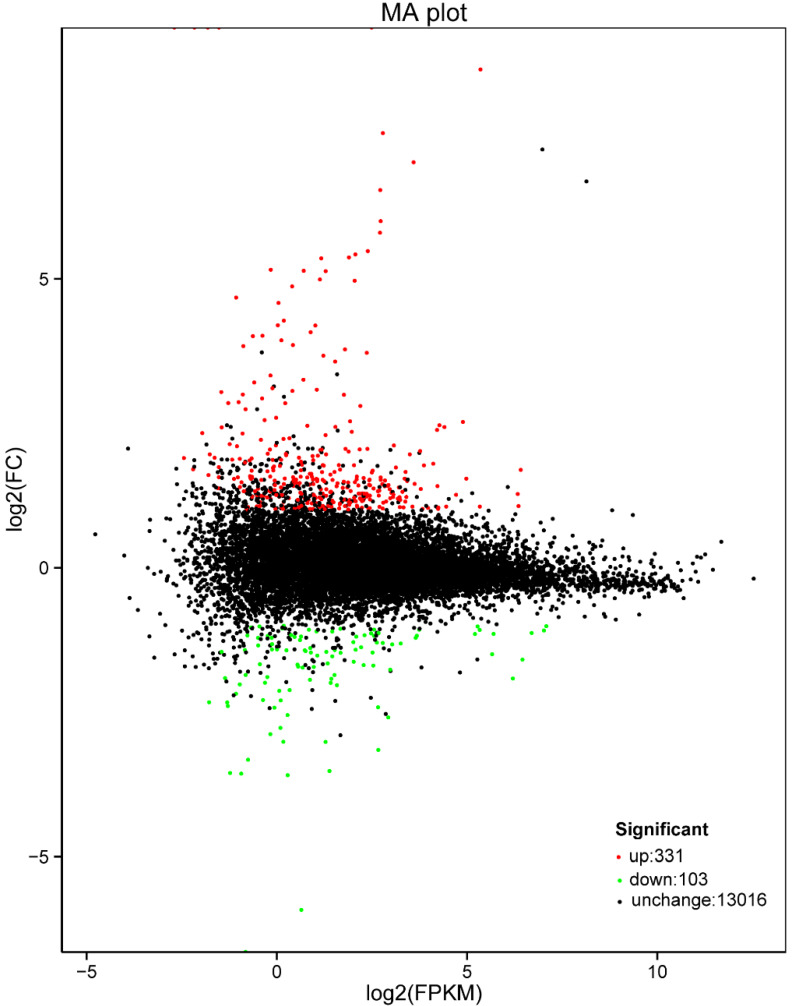
MA plot of the differences in gene expression of mated females relative to gene expression of virgin females.

**Figure 2 insects-11-00308-f002:**
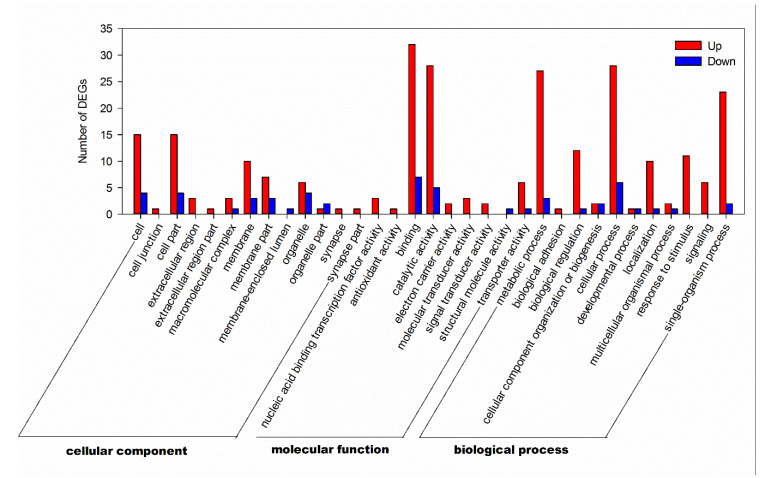
Distribution of up- and down-regulated DEGs among the Gene Ontology (GO) terms in the biological process, cellular component, and molecular function categories.

**Figure 3 insects-11-00308-f003:**
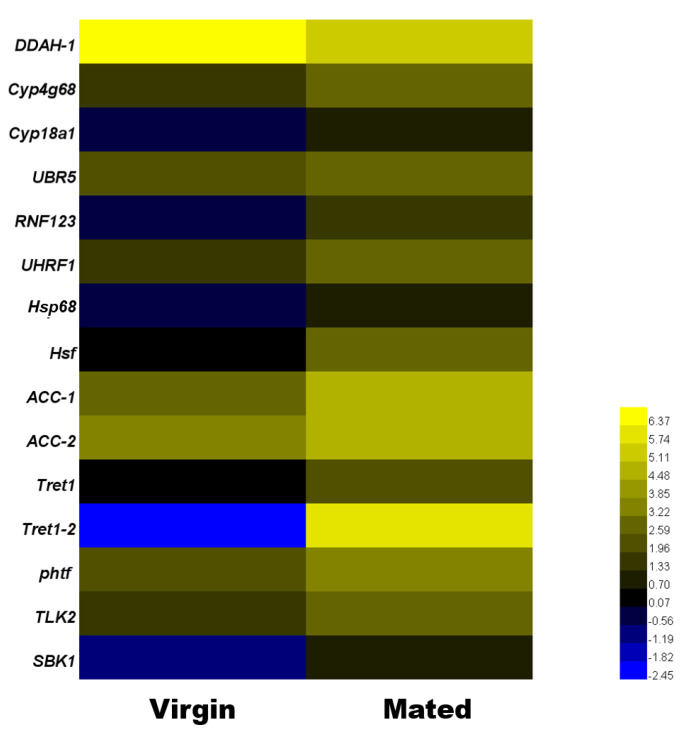
Expression profile of transcripts that are differentially expressed between mated and virgin *B. tabaci* females by RNA-seq. The heatmap shows the transcriptome data of the selected genes, which are based on the log2(fragments per kb of transcript per million fragments mapped (FPKM)) of genes in virgin and mated females. The color scale represents the median-centered log2(FPKM).

**Figure 4 insects-11-00308-f004:**
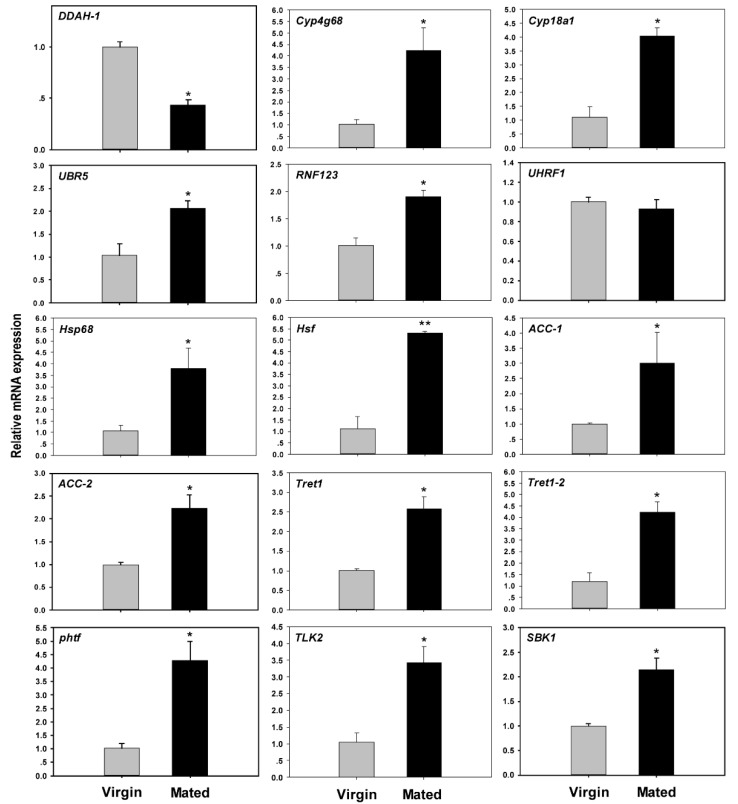
Expression profile of transcripts that are differentially expressed between mated and virgin *B. tabaci* females by RT-qPCR. Mean expression levels (± SEM) of selected genes in virgin females (grey bars) and mated females (black bars). * *p* < 0.05; ** *p* < 0.01.

**Figure 5 insects-11-00308-f005:**
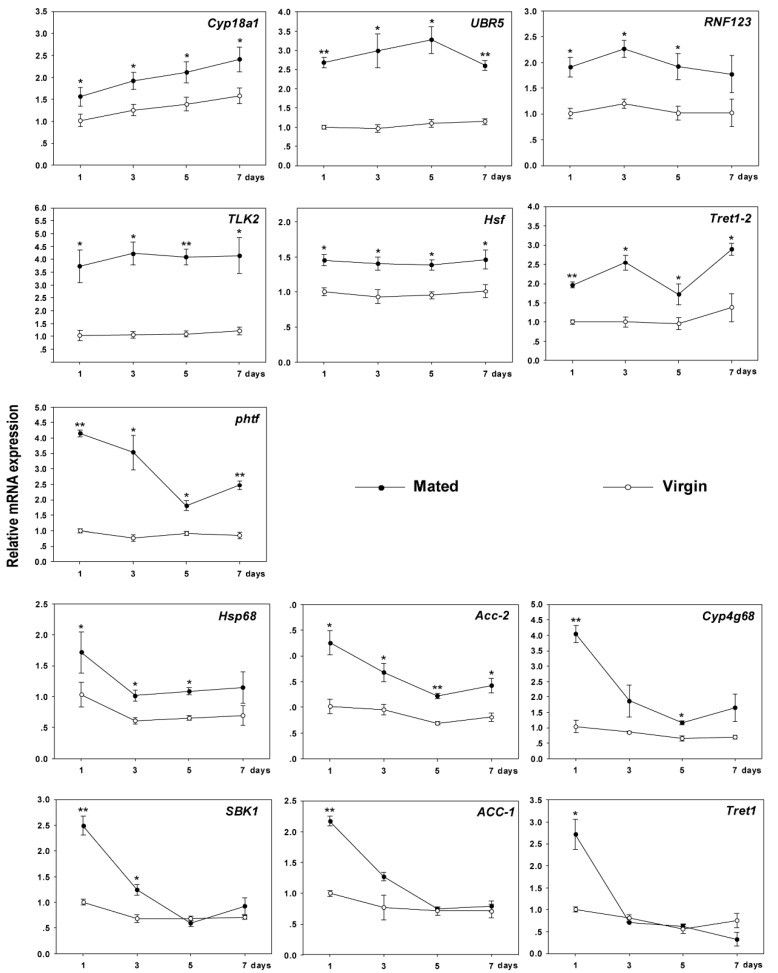
Gene expression profiles of selected mating-related DEGs at different time points for mated and virgin *B. tabaci* females, as determined by RT-qPCR. Mean expression levels (±SEM) of genes in virgin females. Asterisks indicate significant differences between the expression profiles of mated and virgin females. * *p* < 0.05; ** *p* < 0.01.

**Table 1 insects-11-00308-t001:** Statistics of the differentially expressed gene (DEG) results.

Databases	Total	COG	GO	KEGG	KOG	NR	Pfam	Swiss-Prot	eggNOG
DEGs	434	-	-	-	-	-	-	-	-
DEGs-annotated	372	111	77	163	206	372	255	218	307
Upregulation	331	-	66	68	-	-	-	-	-
Downregulation	103	-	11	15	-	-	-	-	-

**Table 2 insects-11-00308-t002:** KEGG pathway enrichment of the top 5.

KEGG Pathway	koID	DEGs Numbers	*p*-Value	Corrected *p*-Value
Longevity regulating pathway—Multiple species	ko04213	7	0.0005238	0.0413826
Protein processing in endoplasmic reticulum	ko04141	10	0.0025248	0.1994616
Pyruvate metabolism	ko00620	5	0.0043336	0.3423515
Phosphatidylinositol signaling system	ko04070	5	0.0125282	0.9897258
Endocytosis	ko04144	8	0.0218859	1

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
