# Peer review of "Transcriptomic Analysis of Mating Responses in Bemisia tabaci MED Females"

_insects, 2020, doi:10.3390/insects11050308_

Round 1
Reviewer 1 Report
Title: A Transcriptome-Wide analysis of mating responses in Bemisia tabaci Med Females
Comments to the editor and authors:
In the above study Huo et al attempted to decipher molecular mechanisms that are associated with post-mating in female whilefly, Bemisia tabaci, a well know polyphagous sucking pest of field crops. Although, the authors used robust approaches such as sequencing, assemblage, annotation and followed by RT PCR analysis; I find that the study lacks clarity with its aim. In fact, I also find that there is heavily influenced by the approaches adopted previously for mosquitoes and fruit flies, questioning the innovations and giving me the impression that the authors carried out this work as an add on species work. Introduction chapter written very badly and require professional language editing. For example, the paragraph starts with arguing the importance of B. tabaci in relevant to their ability to develop genetic resistance to neonicotinoids, then straight away the next paragraph jumps to investigations on reproductive systems from there it pounces to discussing physiological differences between male and female insects after mating. I can find several redundant text in delivering the sentences which are raising questions in my mind that these statements were written without conceiving the content of the study (see examples – commented in the pdf document attached). This sections lacks justification, fails to define the importance of study, and distinguish the lacunae in the literature. So overall the chapter fail to impress the reader for continuous reading (see my comments on the paper itself).
In regard to methodology, I agree that authors have done a detailed work; however, the design mostly resemble the work done in other pests – so merit here is the hard work that the team did in the laboratory. While I am acknowledging the hard work involved; as an independent review, it is pressing me to check for plagiarism. Results section was written to the standard, but again, discussion is dragging and not pertinent to the topic. This kind of expected when the authors could not define the aim and objective very clearly in the introduction. I can understand that the authors have to address all the work that they have done in multiple sections in discussions, however, the chapter lacks interpretation and follow a simply citing the observed results in alignment with available literature in sequence. I suggest authors have to re-write both introduction and discussion with focussing key questions that the study is addressing and how it is relevant to pest management. If not, in what context this study provides info indirectly has to be discussed. The heavy content in the discussion and monotonous style of writing without linking specific research insights in the cited literature and special observations in the current work – making the entire paper not suitable for continuous reading. Overall, I find that the research work is voluminous and not justified (in introduction) and not presented specific insights (in introduction and discussion). Considering this two points, I recommend the editor to reject this paper for publication in Insects.

Author Response
Pont 1: In the above study Huo et al attempted to decipher molecular mechanisms that are associated with post-mating in female whilefly, Bemisia tabaci, a well know polyphagous sucking pest of field crops. Although, the authors used robust approaches such as sequencing, assemblage, annotation and followed by RT-PCR analysis; I find that the study lacks clarity with its aim. In fact, I also find that there is heavily influenced by the approaches adopted previously for mosquitoes and fruit flies, questioning the innovations and giving me the impression that the authors carried out this work as an add on species work. Introduction chapter written very badly and require professional language editing. For example, the paragraph starts with arguing the importance of B. tabaci in relevant to their ability to develop genetic resistance to neonicotinoids, then straight away the next paragraph jumps to investigations on reproductive systems from there it pounces to discussing physiological differences between male and female insects after mating. I can find several redundant text in delivering the sentences which are raising questions in my mind that these statements were written without conceiving the content of the study (see examples – commented in the pdf document attached). This sections lacks justification, fails to define the importance of study, and distinguish the lacunae in the literature. So overall the chapter fail to impress the reader for continuous reading (see my comments on the paper itself).
Response 1: Thank you very much for your comments and suggestions. we have accepted your advice to rewrite introduction chapter and repolished the manuscript by Genesis Technology Communication, Co, Ltd. B. tabaci is a serious pest in many cropping systems worldwide. Mated females played an important role in their outbreak because only mated females can produce female offsprings. We hope to compare transcriptome data of mated and unmated females to find some mating related genes in post-mating females. Mosquitoes and fruit flies as model organism in insects, lots of information related to mating have been reported based on transcriptome and proteomics. But comparing our study with other species’s, we found some mating-related genes were unique to post-mating whitefly females, which gave us some enlightenments that mating may influence the reproduction of female offspring and decide its sex. This study serves as a basis for our future research on the mating of B. tabaci, by helping us selecting mating-related DEGs. We are ready to select some DEGs for RNAi studies, but unfortunately the COVID-19 outbreak in China has delayed our plans.
Point 2: In regard to methodology, I agree that authors have done a detailed work; however, the design mostly resemble the work done in other pests – so merit here is the hard work that the team did in the laboratory. While I am acknowledging the hard work involved; as an independent review, it is pressing me to check for plagiarism. Results section was written to the standard, but again, discussion is dragging and not pertinent to the topic. This kind of expected when the authors could not define the aim and objective very clearly in the introduction. I can understand that the authors have to address all the work that they have done in multiple sections in discussions, however, the chapter lacks interpretation and follow a simply citing the observed results in alignment with available literature in sequence. I suggest authors have to re-write both introduction and discussion with focussing key questions that the study is addressing and how it is relevant to pest management. If not, in what context this study provides info indirectly has to be discussed. The heavy content in the discussion and monotonous style of writing without linking specific research insights in the cited literature and special observations in the current work – making the entire paper not suitable for continuous reading. Overall, I find that the research work is voluminous and not justified (in introduction) and not presented specific insights (in introduction and discussion). Considering this two points, I recommend the editor to reject this paper for publication in Insects.
Response 2: Thanks for your useful criticism. We have accepted your suggestions to rewrite introduction and discussion. In the new version, we use a more refined language to describe introduction and discussion, which stick to the topic of this article.
Reviewer 2 Report
The research was well developed both from the methodological point of view and from that of the analysis of the results. Very interesting. To be published for sure.

Author Response
Dear reviewer,
Thanks for your Comments!
Best wish!
Your sincerely,
Youjun Zhang
Reviewer 3 Report
The paper of Huo et al. describes the post-mating transcriptome of Bemisia tabaci MED. The authors did RNA-seq analyses comparing six pre- and post-mating samples. They reported a total of 434 differentially expressed genes that were functionally characterized via gene enrichment analyses over different databases. Authors carried out RT-PCR to confirm differences in transcript abundances and evaluated their levels over the different time points. The authors identified genes and pathways supporting reproductive processes that can be used in the future for pest control. Overall, the paper is interesting and well written.
However, there are several points which should be addressed prior to publication:
- Mating lasted 24h. Is it possible that this long time affects in some way the results by adding variability (and noise) in gene expression? Did you observe all matings?
- Related to variability in gene expression, the authors should better describe this experiment is based on RNA-pool sequencing. RNA levels fluctuate leading to variability between single samples and, in turn, intra-group variability. Bioinformatics tools have been developed to use replicates (single samples) in order to correctly handle and model. RNA-pool sequencing should be avoided, preferring the barcode strategy of samples as a comparison between RNA-pool sequencing and single sample and sequencing highlighted that pool-seq is prone to false positives results (for more details: PMC4515013). I understand that such strategy is cost-saving, thus I suggest to state and clearly discuss the pros and cons of the strategy to let the readers know about potential biases.
- Each replicate sample was prepared by pooling 40 samples and sequencing produced 20-25 million reads per samples. What was the choice to pool 40 samples and do sequencing with that performances? Is this sequencing enough to capture that variability coming from all the 40 samples? This is important since the amount of molecules that can be sequenced is finite and the fact of having 20 or 40 samples could bias abundance estimation. I also recommend this paper, PMC6755255, for a different point of view.
- It may sound obvious, but which organs have been used for RNA-seq analyses? In which way they have been handled? Is there any possibility of a contamination coming from different nearby fluids/cells?
- Line 113: state the meaning of “low-quality sequences”
- Line 115, 116 and 119: did you use default parameters with the mentioned tools?
- Line 122-124: within each database, did you pick the hit with the lowest E-value? In case of hits with the same score (the score points out to two different proteins), how did you treat the results? And how did you merge the results coming from similar databases (e.g. NCBI NR and Uniprot)?
- Line 126: did you use some parameters/thresholds when using BLAST2GO (e.g. E-value, database restriction, level in GO hierarchy) ?
- Line 132 (and other lines): p-value: the p is lowercase and italicized
- Line 135-137. Gene enrichment analyses were carried out by using the tools GOseq and KOBAS. Several details are missing. Did you use the annotation sets (GO sets, KEGG pathways) retrieved as described at line 122-127? If yes, what is the size of annotation sets (no. of GO and KEGG) and how many genes do they cover (i.e. the background)? If you did not use those sets, opting for different annotations (as they could be derived from a similar organism), can you describe exactly what you did (and the related sizes etc..)? What was the significance threshold used to declare over-representation (alpha level)? Have you considered multiple-testing correction? It is commonly used, otherwise results are too much speculative. Do the enriched GO and KEGG terms carry at least two input DE genes (with just one gene, it is not a true enrichment). All this points should be considered and results, figures and discussions should be revised in case. Figure 2 and 3 should highlight (with a symbol or something else the significantly over-represented terms). Strictly related to KEGG enrichment, Figure 3 reports pathways like morphine addiction or riboflavin metabolism. Having only one DE gene, pathways like those should be removed. What about their meaning for the analysed species? Is it pertinent to report them? I suggest to report data also as tables (GO/KEGG, no. of input genes, size of the terms, background, p-value, adjusted p-value, list of input DE genes belonging to each enriched term).
- Line 151: What are the columns E and R2 in Table S1?
- Line 162: any FDR or Bonferroni correction? I understand that the sample size is low, but results should discuss that any correction has been addressed.
- Table S4 (and similar tables): to what database do you refer when you say “BTA019003.1”? How can we link this identifier to more common “gene names” or NR/Uniprot entries?
- Table S6 (and similar tables): please give an explanation of the column content (e.g. COG_class: what is R, L, RTKL; or KEGG_annotation: I can see the p-value, the ID of a similar gene in another species, et….)
- Line 197 and similar cases: please avoid to start a sentence with a number.
- Line 236-237: A correlation coefficient between qRT-PCR and RNA-seq should be reported as stated at lines 153-154.
Author Response
Dear reviewer,
Thanks for your comments and suggestions. Please see the attachment.
Best wish!
Yours sincerely,
Youjun zhang

Round 2
Reviewer 1 Report
Title: A Transcriptome-Wide analysis of mating responses in Bemisia tabaci Med Females
Comments to the editor and authors on the revised version R1:
I appreciate that the authors have revised the manuscript in alignment with my major comments. Despite this version reads better, I still find the following major and minor conflicts that are mandatory and need to be carried out prior to the publication of this manuscript in Insects.
- Line 15: Remove the word “both”
- Line 67: Insert a few lines about the application of this work – address directly or indirectly- see my previous revision comment # “focussing key questions that the study is addressing and how it is relevant to pest management. If not, in what context this study provides info indirectly has to be discussed”
- Line 82: unmated groups – are they virgin insects? If so, I would suggest to change all to Virgin groups
- Line 144: What is the basis of selection of a set of specific 15 transcripts from DEGS for qPCR validation? This needs to be clearly explained. This is a primary concern and thus without a proper justification, the manuscript stands a very weak and pave ways for negative comments.
- Line 160-166: This section needs to be summarised as a separate with some key features. For example, a summary of statistic of mated and unmated (virgin) B. tabaci transcriptome sequences to be given
- Line 191-198: Why the authors have selected DEG base on Go terms – instead of using specific pathway analysis. In general, the latter is more appropriate in transcriptome analysis study. Selecting DEGs based on Go terms may be preliminary and even in this case, the authors did not provide specific descriptions such as based on which key processes: biological, molecular and cellular. I strongly suggest to authors to use Pathway analysis on their data and compare with the Go terms and describe whether they have got similar results or different
- Why the authors did not perform sequence homology search (of the key identified genes) within NCBI or relevant databases and found whether similar work was reported with other species of whiteflies (only Aleurodidae) such as spiralling whitefly and blackfly- this is very important and add significance and scope of this study. I strongly suggest to the authors to do this work and interpret it in the discussion. This will also increase the impact of the study especially with the application part of it.
- I knew that authors have indicated the outbreak of COVID-19 for not carrying out laboratory analysis. However, comments 5 and 6 – doesn’t require any laboratory work and they can be done online with the basic power computing systems.
- Line 246-248: Rewrite the lines – it is confusing
- Figure 5. In this graph, What is the symbol of URB5 refers to? I suppose this is UBR5- clarify. If I am correct – manuscript needs to be checked thoroughly for such typos and formatting error.
- Title: the phrase MED on title needs to be expanded
- Introduction and Discussion: Revise it in accordance with three of major comments above: #1, #3 and #6. This is compulsory and compensating on these may lead to the rejection of this paper
Author Response
Dear reviewer,
Thanks for your comments on our manuscript, we have studied comments carefully and have made correction which we hope meet with approval.
Point 1: Line 15: Remove the word “both”.
Response 1: Thanks for your suggestion, I have removed the word “both ” in line 15.
Point 2: Line 67: Insert a few lines about the application of this work – address directly or indirectly- see my previous revision comment “focussing key questions that the study is addressing and how it is relevant to pest management. If not, in what context this study provides info indirectly has to be discussed”.
Response 2: Thanks for your advice. For this study, we want to find some genes which response to mating, and verified some key genes in different mating time points. In future, we hope to reduce the success rate of fertilization by blocking the expression of these key genes in mated females, which will reduce female offspring to decline the population density of this pest. Because for Bemisia tabaci, only mated females can produce female offspring. We have made some supplements in lines 66-67 in the introduction part and 345-348 in the discussion part.
Point 3: Line 82: unmated groups – are they virgin insects? If so, I would suggest to change all to Virgin groups.
Response 3: As you said, the “unmated group” stands for virgin insects. And we have accepted your suggestion to change all of them to “virgin groups or virgin females”.
Point 4: What is the basis of selection of a set of specific 15 transcripts from DEGS for qPCR validation? This needs to be clearly explained. This is a primary concern and thus without a proper justification, the manuscript stands a very weak and pave ways for negative comments.
Response 4: Before we selected genes to verify, we read lots of references about responsing to mating in other insects. Based on the reported genes family and metabolic pathways, we selected 15 DEGs, and we have described carefully in lines 145-148 and 235.
Point 5: Line 160-166: This section needs to be summarised as a separate with some key features. For example, a summary of statistic of mated and unmated (virgin) B. tabaci transcriptome sequences to be given.
Response 5: Thanks for your attention. In fact, we have submitted all transcriptome sequences to the Bioproject accession number PRJNA559034. And a summary of statistic of mated and virgin B. tabaci transcriptome sequencing information was given in the supplementary Table S2 in order to save space.
Point 6: Line 191-198: Why the authors have selected DEG base on Go terms – instead of using specific pathway analysis. In general, the latter is more appropriate in transcriptome analysis study. Selecting DEGs based on Go terms may be preliminary and even in this case, the authors did not provide specific descriptions such as based on which key processes: biological, molecular and cellular. I strongly suggest to authors to use Pathway analysis on their data and compare with the Go terms and describe whether they have got similar results or different.
Response 6: Thanks for your good suggestions. We have supplementary specific descriptions for those DEGs, which belonged to biological process, molecular function or cellular component in lines 194-200. In addition, we also described DEGs based on KEGG pathway in lines 205-216. Furthermore, we have accepted your suggestion to use Pathway analysis on their data and compare with the Go terms in lines 219-227.
Point 7: Why the authors did not perform sequence homology search (of the key identified genes) within NCBI or relevant databases and found whether similar work was reported with other species of whiteflies (only Aleurodidae) such as spiralling whitefly and blackfly - this is very important and add significance and scope of this study. I strongly suggest to the authors to do this work and interpret it in the discussion. This will also increase the impact of the study especially with the application part of it.
Response 7: Thanks for your suggestions. Before we designed this exprement, we had tried to find other reports on the response of mating in other species of whiteflies, especially at the molecular level. However, it is a pity that there is no report in other species of whiteflies. We only find those we have cited species shuch as Drosophila melanogaster, Apis mellifera, Ceratitis capitate and so on. In this time, we try our best to find in other species of whiteflies about mating response reports again, but there is still not. I hope my work can fill this gap.
Point 8: I knew that authors have indicated the outbreak of COVID-19 for not carrying out laboratory analysis. However, comments 5 and 6 – doesn’t require any laboratory work and they can be done online with the basic power computing systems.
Response 8: Thanks for your attention, I have accepted your suggestions to supplement those contents.
Point 9: Line 246-248: Rewrite the lines – it is confusing.
Response 9: Thanks for your advice, I have rewrited that sentence in lines 259-261.
Point 10: Figure 5. In this graph, What is the symbol of URB5 refers to? I suppose this is UBR5- clarify. If I am correct – manuscript needs to be checked thoroughly for such typos and formatting error.
Response 10: Thanks for your attention. We are sorry to have made a mistke in spelling. We have checked thoroughly. And new figures have been submitted.
Point 11: Title: the phrase MED on title needs to be expanded.
Response 11: “Bemisia tabaci MED” is a biotype of Bemisia tabaci, which contained many biotypes. This named is a equial to Bemisia tabaci Q.
Point 12: Introduction and Discussion: Revise it in accordance with three of major comments above: #1, #3 and #6. This is compulsory and compensating on these may lead to the rejection of this paper.
Response 12: Thanks for your comments on our manuscript. We have studied 1st, 3rd and 6th comments carefully and have made correction which we hope meet with approval.
Best wish!
Yours sincerely,
Youjun Zhang
Reviewer 3 Report
Dear authors,
thanks for having considered the suggested changes and improved the manuscript.
Author Response
Dear reviewer,
Thanks for your comments on our manuscript.
Best wish!
Yours sincerely,
Youjun Zhang
Round 3
Reviewer 1 Report
Thanks for understanding my comments. But I still find minor errors and major conflicts in interpreting the scientific findings. I strongly recommend to carefully write these results without any mistakes. Language delivery is erroneous and confusing. For example - the statements are hanging and starting with the word "And" and numerals "15". I am also advise to write the reasoning for the selection of 15 DEGs in detail meticulously. As of now, it just fragmented and not clear and not connected. Thirdly, writing application aspect of the research doesn't mean writing something which cant be achievable and not pragmatic. Gene suppression that the authors mentioned is not perpetual and not practical use. I have come up with the hint based on your work (see the MS for my comments) but it is authors response to elaborate this.
I feel the authors in a hurry to address all of my comments and added more errors to the manuscript. I request the authors to take 2-3 weeks time to work on this and make sure everything is error-free and not confusing.

Author Response
Dear reviewer,
Thanks for your comments. We have studied comments carefully and have made correction which we hope meet with approval.
Point 1: Thanks for understanding my comments. But I still find minor errors and major conflicts in interpreting the scientific findings. I strongly recommend to carefully write these results without any mistakes. Language delivery is erroneous and confusing. For example - the statements are hanging and starting with the word "And" and numerals "15".
Response 1: Thanks for your advice, I have checked all manuscript and rewritten those by a native English speaker.
Point 2: I am also advice to write the reasoning for the selection of 15 DEGs in detail meticulously. As of now, it just fragmented and not clear and not connected.
Response 2: Thanks for your attention. Fifteen DEGs, which were used to qRT-PCR validation, contained three ubiquitin-protein ligase genes (UBR5, UHRF1 and RNF123), two cytochrome P450 genes (Cyp4g68 and Cyp18a1), two heat-shock protein genes (Hsp68 and Hsf), two facilitated trehalose transporter genes (Tret1 and Tret1-2), two serine-protein kinase genes (TLK2 and SBK1), two carboxylase genes (ACC-1 and ACC-2), transcription factor (phtf) and hydrolase (DDAH-1). The main reasons why these genes were selected are following: The family of cytochrome P450 genes had been reported in the mating females of Aedes aegypti and Drosophila melanogaster [1,3,4,7,8]. The homology of ubiquitin-protein ligase genes had been reported in post-mating Apis mellifera and A. aegypti females [2,7]. The homology of serine/threonine-protein kinase (TLK2 and SBK1) were reported in mating females of A. mellifera, A. aegypti and D. melanogaster [2,7,8]. Hsp genes were reported in the mating females of D. melanogaster and A. aegypti [1,6,7]. The homology of transcription factor and trehalose transporter (Tret1 and Tret1-2) were reported in post-mating A. aegypti females [7]. Carboxylases (ACC-1 and ACC-2) and hydrolases (DDHA-1) were unique to the post-mating B. tabaci females. We briefly summarize in lines 138-141.
- Lawniczak, M. K.; Begun, D. J. A genome-wide analysis of courting and mating responses in Drosophila melanogaster Genome. 2004, 47, 900-910.
- Kocher, S. D.; Richard, F. J.; Tarpy, D. R.; Grozinger, C. M. Genomic analysis of post-mating changes in the honey bee queen (Apis mellifera). BMC Genomics. 2008, 9, 232.
- Rogers, D. W.; Whitten, M. M.; Thailayil, J.; Soichot, J.; Levashina, E. A.; Catteruccia, F. Molecular and cellular components of the mating machinery in Anopheles gambiae Proc. Natl. Acad. Sci. U. S. A. 2008, 105, 19390-19395.
- Innocenti, P.; Morrow, E. H. Immunogenic males: a genome-wide analysis of reproduction and the cost of mating in Drosophila melanogaster J. Evol. Biol. 2009, 22, 964-973.
- Gomulski, L. M.; Dimopoulos, G.; Xi, Z.; Scolari, F.; Gabrieli, P.; Siciliano, P.; Clarke, A. R.; Malacrida, A. R.; Gasperi, G. Transcriptome profiling of sexual maturation and mating in the Mediterranean fruit fly, Ceratitis capitata. PloS one. 2012, 7, e30857.
- Short, S. M.; Lazzaro, B. P. Reproductive status alters transcriptomic response to infection in female Drosophila melanogaster. G3: Genes, Genomes, Genetics. 2013, 3, 827-840.
- Alfonso-Parra, C.; Ahmed-Braimah, Y. H.; Degner, E. C.; Avila, F. W.; Villarreal, S. M.; Pleiss, J. A.; Wolfner, M. F.; Harrington, M. F.; Harrington, L. C. Mating-induced transcriptome changes in the reproductive tract of female Aedes aegypti. Plos Neglect. Trop. Dis. 2016, 10, e0004451.
- Kapelnikov, A.; Zelinger, E.; Gottlieb, Y.; Rhrissorrakrai, K.; Gunsalus, K. C.; Heifetz, Y. Mating induces an immune response and developmental switch in the Drosophila Proc. Natl. Acad. Sci. U. S. A. 2008, 105, 13912-13917.
Pont 3: Thirdly, writing application aspect of the research doesn't mean writing something which can’t be achievable and not pragmatic. Gene suppression that the authors mentioned is not perpetual and not practical use. I have come up with the hint based on your work (is this a perpetual application.. i doubt and practically not possible? The application may not direct -- it should arise based on the detailed knowledge and understanding of the proposed research topic. This demonstrates that the authors did not have detailed knowledge on application aspect of pest management. I have given my opinion here: The research in fact reveals strong biochemical insights that happening within female flies post mating. More investigation on physiological effects of the discussed DEGs or pathways may faclitate key pathways that are involved in fertilization and embryo delvelopment (egg laying). This special insights may be exploited for managing this pest in the future.) but it is authors response to elaborate this.
Response 3: Thanks for your comments. We have accepted your opinion and deleted the relevant sentence.
Best wish!
Yours sincerely,
Youjun Zhang